# Acquired Angioedema in Selected Neoplastic Diseases

**DOI:** 10.3390/medicina59040644

**Published:** 2023-03-24

**Authors:** Magdalena Zając, Andrzej Bożek, Renata Kozłowska, Alicja Grzanka

**Affiliations:** Clinical Department of Internal Diseases, Dermatology and Allergology, Medical University of Silesia, 40-055 Katowice, Poland

**Keywords:** angioedema, IgE, neoplasma, allergy

## Abstract

*Background and Objectives*. Acquired angioedema is a relatively common revelation accompanying some diseases such as autoimmune or cancer. The study aimed to assess the incidence of one subtype of angioedema—C1-INH-AAE (acquired angioedema with C1 inhibitor deficiency). *Material and methods*. The study was retrospective and based on 1 312 patients with a final diagnosis of breast cancer, colorectal cancer, or lung cancer: 723 women and 589 men with a mean age of 58.2 ± 13.5 years. The cancer diagnosis according to the ICD (International Classification of Diseases)-10 code, medical history including TNM (Tumour, Node, Metastasis) staging, histopathology, and assessment of the occurrence of C1-INH-AAE angioedema were analysed. *Results*. C1-INH-AAE occurred more often in patients with cancer than in the control group, as follows: 327 (29%) vs. 53 (6%) for *p* < 0.05. C1-INH-AAEs were observed most often in the group of patients diagnosed with breast cancer compared to colorectal and lung groups: 197 (37%) vs. 108 (26%) vs. 22 (16%) (*p* < 0.05). A higher incidence of C1-INH-AAE was observed in the early stages of breast cancer. However, there was no relationship between the occurrence of C1-INH-AAE and the BRCA1 (Breast Cancer gene 1)/BRCA2 (Breast Cancer gene 2) mutation or histopathological types of breast cancer. *Conclusion*. Angioedema type C1-INH-AAE occurs more often in patients with selected neoplastic diseases, especially in the early stages of breast cancer.

## 1. Introduction

Angioedema is a clinical entity defined as self-limiting oedema that is localised in the deeper layers of the skin and mucosa and lasts for several days [1,2,3].

Angioedema is one of the common skin diseases. It can coexist with urticaria or be present alone.

The prevalence of angioedema is estimated to be between 2 and 20% of the population. It depends on the way of analysis, for example, with or without concomitant urticaria. However, angioedema is also considered as one of the forms of urticaria [4,5,6,7]. According to the European Academy of Allergy and Clinical Immunology classification, angioedema without wheels as a separate disease is classified based on its cause, acquired or hereditary, and response to treatment. Four types of acquired (AAE) and three types of hereditary (HAE) angioedema are identified [1,3,4]. Another similar classification of angioedema is based on the kinds of mediators responsible for symptom development, including bradykinin-induced angioedema (AE), mast cell mediator-induced AE, and AE due to an unknown mediator [1,3]. For example, uncontrolled production of bradykinin (BK) due to insufficient levels of protease inhibitors controlling contact phase (CP) activation, increased activity of CP proteins, and/or inadequate degradation of BK to inactive peptides increases vascular permeability via the BK receptor 2 (BKR2) and causes subcutaneous and submucosal oedema [8,9]. On the other hand, in mast cells, mediator-induced angioedema, recruitment of inflammatory cells to skin mast cell activation, and degranulation sites contribute to angioedema via histamine and other mediators [10,11,12,13].

AAE episodes may have various triggers, such as mild trauma, cold exposure, viral or bacterial infections, pregnancy, stress, certain foods, and chronic auto-immunological or neoplastic diseases [1,14]. The incidence of angioedema episodes is unpredictable [1].

The most common form of angioedema associated with other systemic diseases is C1-INH-AAE. It is considered a subset of AAE. This form of this AAE is characterised by the acquired consumption of C1-INH associated with different systemic disorders. The negative family history for angioedema, age of onset > 40 years, acquired deficiency of C1-INH, hyperactivation of the classic complement pathway, and recurrent angioedema episodes are characteristic of C1-INH-AAE [1,15]. This type is most commonly related to B-cell lymphoproliferative diseases. Other reported neoplastic disorders for C1-INH-AAE are T-cell lymphoma, multiple myeloma, chronic lymphocytic leukaemia, rectal carcinoma, and non-Hodgkin lymphoma [1,14,15]. The involvement of other types of neoplasms is possible. The associated risk for haematolymphoid malignancy is 35%, and for other malignancies, 8%, based on a study of 128 patients [16].

The study aimed to assess the possible impact of the selected most common malignancies on the risk of C1-INH-AAE. An additional aim of the study was to evaluate the effect of breast cancer stage and BCR1 and BCR2 gene mutations on the risk of this type of angioedema.

## 2. Material and Methods

The study was retrospective and based on the study group consisting of people with a final diagnosis of breast cancer, colorectal cancer, or lung cancer, and a history of suspected angioedema (according to International Classification of Diseases (ICD) code 10) in the last year. According to data, these malignant tumours are the most common in Poland [17]. The control group consisted of people who had not been diagnosed with cancer, and who had a history of suspected angioedema in the last year. The study included 1312 patients: 723 women and 589 men aged 35 to 75, with a mean age of 58.2 ± 13.5 years. The examined patients were diagnosed in oncology clinics and had a final diagnosis of cancer based on histopathology, but before primary oncology treatment. The group was selected by analysing 11,274 records and sample stratification, the key being age 35–70. This age range was selected based on the average age median of patients suffering from these cancers according to Polish statistical data [17].

### 2.1. Inclusion Criteria for the Study

age between 35–70 years;confirmed breast, colorectal or lung cancer before starting targeted treatment;negative family history for angioedema, acquired deficiency of C1-INH, which could suggest C1INH-AEE angioedema.

### 2.2. Exclusion Criteria from the Study

The exclusion criteria were angioedema with chronic urticaria, mild angioedema-like symptoms such as swelling of the tongue or mouth without visible oedema, documented angioedema induced by drugs (for example analgesics: nonsteroidal anti-inflammatory drugs, beta lactam antibiotics), use of convertase inhibitors or their derivatives, oedema reactions that could not be clearly classified as angioedema, deficiencies in documentation, and lack of consent. However, patients whose discontinuation of convertase inhibitors or other suspected drugs did not resolve the oedema symptoms were enrolled in the study.

Additionally, hereditary angioedema was also checked based on the ICD-10 code and excluded from the study with no mutations of the SERPING1 gene, and also as a clinical diagnosis which was based on the following medical history points: (1) a positive family history; (2) onset of symptoms in childhood/adolescence; (3) painful abdominal symptoms; (4) occurrence of upper airway oedema; (5) lack of response to antihistamines, glucocorticoids, or epinephrine; (6) presence of prodromal signs or symptoms before swelling; and/or (7) absence of urticaria (4).

The control group was 1450 people selected from Family Doctor Outpatient Clinics. This group was matched in terms of age and sex, comparable to the patients in the study group. A total of 935 patients were finally obtained, including 471 women and 474 men aged 35 to 75, with an average age of 55.9 ± 9.5 years.

### 2.3. Research Methodology

All enrolled patients underwent a retrospective assessment for

diagnosis of cancer according to the IC-10 code, medical history including TNM staging, histopathology, and other documentation (CT, MR);assessment of the occurrence of angioedema incident (ICD-10 code, medical history and documentation) confirmed as C1-INH-AAE with decreased serum C1-INH.

The project was approved by the Bioethical Committee of the Medical University of Silesia in Katowice, Poland (no. KNW/0022/KB1/18/14, approval date 10 May 2020).

### 2.4. Statistics

Statistica software Version 8.2 (Statsoft, Cracow, Poland) was used for calculations. The significance of differences for normal distributions was assessed with the Student’s *t*-test and for non-normal distributions with the U-Mann–Whitney test. The frequency of the observed feature was calculated, and the percentages were compared using the chi-square test. The Pearson test was used to assess the correlation. *p* < 0.05 was considered statistically significant.

## 3. Results

Finally, 1115 patients with cancer and 843 controls were analysed. The characteristics of the patients are presented in Table 1. C1-INH-AAE occurred more often in patients with cancer than in the control group as follows: 327 (29%) vs. 53 (6%) for *p* < 0.05.

At the same time, C1-INH-AAEs occurred most often in the group of patients diagnosed with breast cancer compared to colorectal and lung groups: 197 (37%) vs. 108 (26%) vs. 22 (16%) (*p* < 0.05). This was related to a significantly lower mean C1-INH concentration in this group (Figure 1). However, the functional activity of C1-INH was comparable in the studied cancer subgroups and the control group (Figure 2). The mean complement C4 concentration was similar in the cancer subgroups studied, and significantly lower than in the control group (Figure 3).

### Sub-Analysis of Patients with Breast Cancer

Due to the dominance of AAE in the group of patients diagnosed with breast cancer, this group was subjected to additional detailed analysis.

In this subgroup, a higher incidence of C1-INH-AAE was observed in the early stages of the disease (Figure 4). There was no relationship between the occurrence of AAE and the BRCA1/BRCA2 mutation or histopathological types of breast cancer (Table 2). However, an increase in BMI resulted in a higher incidence of C1-INH-AAE in the subjects with breast carcinoma.

A thorough analysis of the bowel and lung cancer groups was not performed due to the lack of precise specific data. 

## 4. Discussion

Angioedema due to C1 esterase inhibitor (C1-INH) deficiency is one of the leading causes of bradykinin-mediated angioedema [1]. Testing for C1-INH deficiency is performed by measuring C1-INH antigen level, C1-INH function, and C4 levels in plasma. C1-INH deficiency can be due to a genetic defect (hereditary angioedema, HAE) or an acquired defect [15]. Acquired angioedema is a possible symptom in autoimmune and cancer diseases [1,15], but its causes are still not precisely defined. Autoimmune diseases are usually less frequently reported in C1-INH-AAE. The predominant disease appears to be autoimmune thyroiditis, although it is generally not a common finding. Occasional cases of lupus and rheumatoid arthritis are reported. It is worth noting that many patients with autoimmune disease had concomitant haematological disorders, such as leukopenia, anaemia, and secondary cellular and humoral immunodeficiencies [18,19,20].

In the presented work, we focused only on C1-INH-AAE as it is prevalent in neoplastic diseases [1,3]. In the proposed observation, the occurrence of angioedema incidents meets the definitions of C1-INH-AAE, i.e., acquired deficiency of C1-INH, hyperactivation of the classic pathway of human complement, and recurrent angioedema symptoms.

The results confirmed a relatively frequent form of angioedema in neoplastic diseases. However, other authors observed the presence of C1-INH-AAE mainly in myeloproliferative disorders, especially in selected forms of leukaemia [2,21]. This observation presents a more frequent form of angioedema in other neoplastic diseases. The reported incidence of C1-INH-AEE in breast, bronchial, or lung cancer is the first such extensive observation, apart from single case reports [22,23,24]. 

The neoplasm disease may produce idiotype or anti-idiotype antibodies or immune complexes that decrease the concentration or activity of C1-INH. In our study, this first event was observed. Consequences of increased consumption of C1q followed by C2 and C4 result in the subsequent release of vasoactive peptides that act on postcapillary venules. In a few studies, the mechanism by which auto-antibodies consume C1-inhibitor was checked. These observations suggest that these antibodies bind epitopes around the reactive centre of the C1-inhibitor [25]. This binding could either create a steric impairment to the formation of C1-inhibitor–protease complexes or convert C1-inhibitor into a substrate for proteases, accounting for large amounts of a cleaved inactive form of C1-inhibitor that is detected in the serum of most patients with acquired C1-inhibitor deficiency [26]. Some authors think that another probable explanation for the development of angioedema is the overconsumption of C1-INH by neoplastic tissue [1,2]. Therefore, there was also a historical classification of C1-INH-AAE into type I (absence of C1-INH autoantibodies) and type II (presence of C1-INH autoantibodies). However, in this study, we did not separately analyse those types. The pathophysiology of C1-INH-AEE in breast cancer is not so clear. However, the proposed mechanism of overconsumption of C1inh by neoplastic lymphatic tissue in breast cancer seems decisive. The growth of neoplastic lymphoid tissue in the first phase of this disease may further explain the increased consumption of C1- INH in the described mechanism, resulting in a higher frequency of oedema. 

Geha et al. confirmed evidence that the mechanism of C1-INH consumption in B-cell lymphoproliferative disorders was caused by the formation of idiotype/anti-idiotype immune complexes [27]. Other authors also observed an autoreactive immunoglobulin G against C1-INH, being evidence that some cases of C1-INH-AAE could have an autoimmune basis [28]. There is also an animal model observation which investigated the properties of splenic lymphosarcoma tissue resected from a patient with AAE and splenic lymphosarcoma, and demonstrated that in vitro incubation of lymphosarcoma tissue with guinea pig serum led to complement depletion [29]. Similar evidence studies at the molecular level should therefore be carried out in other neoplastic diseases, especially in presented breast cancer, to finally confirm the proposed mechanisms of angioedema induction. This requires further research.

When analysing the subgroup of patients with breast cancer, it is worth emphasising the increase in the frequency of swelling in the early stages of the disease, which may be explained by the increased activity of the described immune mechanisms involving complement. This phenomenon should induce oncological vigilance in these patients and make appropriate diagnostics.

On the other hand, the lack of correlation between the occurrence of C1-INH-AAE and histopathological types of cancer or BRCA mutations shows a specific independence of this symptom and a universal character. However, this requires further research. In the past, attempts have been made to link the appearance of angioedema to specific mutations in lymphoproliferative neoplasms, but these have not been confirmed [30].

In the described cases of acute anaplastic leukaemia and non-Hodgkin lymphoma, angioedema was observed as an early sign of neoplastic disease, similar to the presented results concerning breast cancer. However, that study reported only isolated cases [31].

The risk of breast cancer increases by 12% for every 5 kg/m^2^ above average body weight. In women with a BMI over 28 kg/m^2^, the risk increases by 26%. In addition, obesity is a modifiable risk factor for developing breast cancer. In the presented study, most women with breast cancer had elevated BMI, especially in the early phase of the disease. It could explain the relationship between BMI and the occurrence of angioedema [32,33]. However, why this is related to C1-INH-AEE requires further research.

The presented work has its limitations. Apart from the ones shown earlier, it is essential that other types of angioedema cannot be excluded entirely in the study group, especially the congenital form. Despite restrictive methods of qualification, mutation assessment, and the more advanced age of patients, the involvement of HAE cannot be ruled out. The limitation is also the lack of determination of anti-C1-INH antibodies.

Patients with a standard C1-INH value and oedema were not analysed in the study to maintain the validity of the observations. They were not analysed due to their small number. It is worth emphasising, however, that other forms of angioedema were incidental in these patients in single cases.

## 5. Conclusions

Angioedema type C1-INH-AAE occurs more often in patients with selected neoplastic diseases, especially in the early stages of breast cancer. The phenomenon requires further research. 

## Figures and Tables

**Figure 1 medicina-59-00644-f001:**
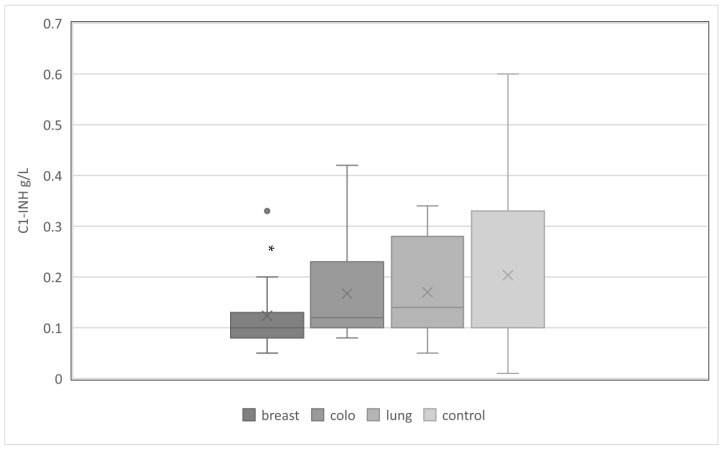
The serum concentration of C1-INH in study groups with C1-INH-AAE. Legend: C1-INH: C1 inhibitor; breast—breast cancer; colo—colorectal cancer; lung—lung cancer; *—significantly lower average concentration of C1-INH in patients with breast cancer compared to other groups, *p* < 0.05.

**Figure 2 medicina-59-00644-f002:**
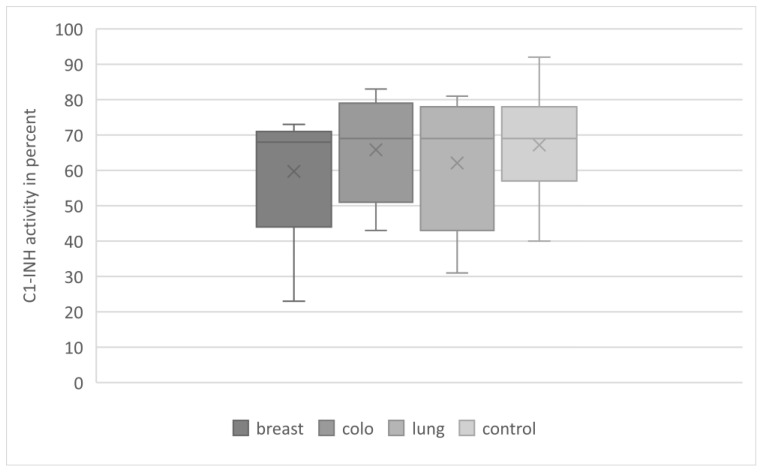
C1-INH activity in study groups with C1-INH-AAE. Legend: C1-INH: C1 inhibitor; breast—breast cancer; colo—colorectal cancer; lung—lung cancer.

**Figure 3 medicina-59-00644-f003:**
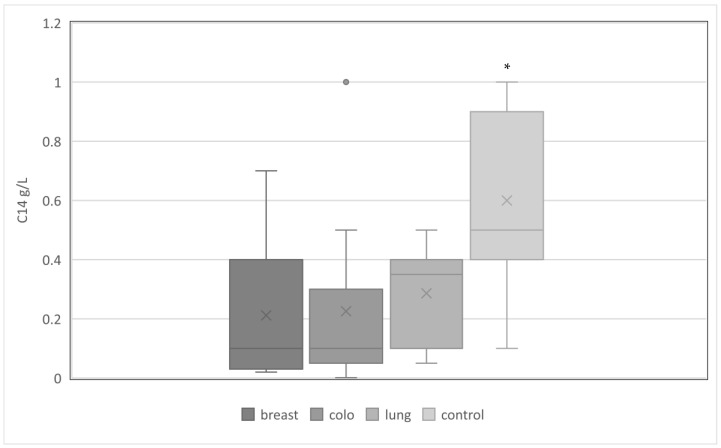
The serum concentrations of C4 in studied groups with C1-INH-AAE. Legend: breast—breast cancer; colo—colorectal cancer; lung—lung cancer; *—significantly higher average concentration of C4 in control groups, *p* < 0.05.

**Figure 4 medicina-59-00644-f004:**
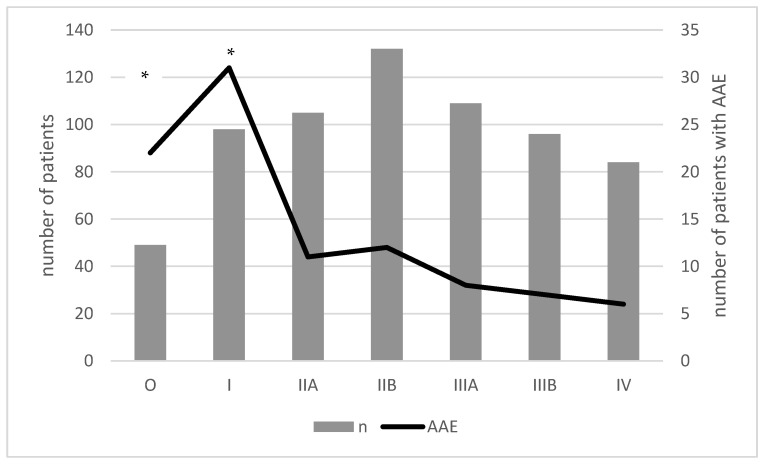
The incidence of C1-INH-AAE in patients with breast carcinoma according to TMN classifications. Legend: Stage 0: Tis; Stage I: T1N0; Stage IIA: T2N0, T3N0 T0N1; Stage IIA: T1N1, T2N1; Stage III: Invasion into skin and/or ribs, matted lymph nodes, T3N1, T0N2, T1N2, T2N2, T3N2, AnyT N3, T4 any N, locally advanced breast cancer; Stage IV: M1, advanced breast cancer; *—significant higher number of patients with incidence of C1-INH-AAE in 0 and I stage of TMN.

**Table 1 medicina-59-00644-t001:** Characteristics of the study and control groups.

	Cancer Group *n* = 1115	Control Group *n* = 843	*p*
mean age in years	58.2 ± 13.5	62.8 ± 10.6	>0.05
mean BMI (kg/m^2^) ± SD	22 ± 7	25 ± 8	>0.05
men (%)	779 (70)	562 (67)	>0.05
urban area (%)	856 (76)	649 (77)	>0.05
smoker or former smoker (%)	571 (40)	399 (47)	>0.05
malignant cancer (%) breast colorectal lung	565 (50) 410 (37) 140 (13)	- - -	- - -
number of angioedema incidences (%): one two to three four or more (recurrent)	327 (29) 127 (11) 103 (9) 97 (8)	53 (6) 34 (64) 11 (21) 8 (15)	0.01 0.01 0.01 0.03
hospitalization due to angioedema (%)	93 (8)	10 (19)	0.02
main localization (%) head throat hands or feet multi-location	*n* = 327 145 (44) 43(13) 60 (18) 79 (24)	*n* = 53 26 (49) 10 (19) 12(23) 5 (9)	>0.05 0.04 0.03 0.01

Legend: SD—standard deviation, BMI—body mass index.

**Table 2 medicina-59-00644-t002:** The impact of the studied variables on the occurrence of C1-INH-AAE.

Patients with Breast Cancer		
Variables	Pearson’s Correlation r_s_ with Presence of C1-INH-AAE	*p*
BMI	0.856	0.012
Smoking	0.494	0.083
BRCA1 mutation	0.032	0.569
BRCA2 mutation	0.028	0.713
C15.3	−0.113	0.105
Ductal carcinoma in situ Lobular carcinoma in situ Invasive ductal carc. Invasive lobular carc. Adenocystic carcinoma Metaplastic carcinoma Micropapillary carc Mucinous carcinoma Papillary carcinoma Tubular carcinoma Mixed carcinoma	0.213 0.098 −0.231 0.119 0.102 −0.099 −0.345 −0.11 0.082 0.028 0.201	0.231 0.732 0.118 0.452 0.221 0.879 0.102 0.889 0.131 0.229 0.093

## Data Availability

The data presented in this study are available on request from the corresponding author. The data are not publicly available due to ethical restrictions.

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
