# Peer review of "Acquired Angioedema in Selected Neoplastic Diseases"

_medicina, 2023, doi:10.3390/medicina59040644_

Round 1
Reviewer 1 Report
Authors described that C1-INH-AAE is characterized by acquired deficiency of C1- INH. Did authors use serum level of C1-INH for diagnosis in this study? If not, why?
What is the reason why C1-INH- AAEs occurred most often in the group of patients with breast cancer?
Are there any evidence showing the increased activity of the immune mechanisms involving complement in early stage compared to late stage cancer?
In other cancer such as colorectal and lung cancer, are C1-INH-AAEs more frequent in early stage?
Why is BMI associated with occurrence of C1-INH-AAEs in breast cancer? Is BMI also associated with occurrence of C1-INH-AAEs in other cancer except for breast cancer?
Author Response
Authors described that C1-INH-AAE is characterized by acquired deficiency of C1- INH. Did authors use serum level of C1-INH for diagnosis in this study? If not, why?
Answer. Yes. We used serum levels of C1-INH for diagnosis. We added information about it in the methods.
What is the reason why C1-INH- AAEs occurred most often in the group of patients with breast cancer?
Answer. The pathophysiology of C1-INH-AEE in breast cancer is not so clear. However, the proposed mechanism of overconsumption of C1inh by neoplastic lymphatic tissue in breast cancer seems decisive. The growth of neoplastic lymphoid tissue in the first phase of this disease may further explain the increased consumption of C1INH in the described mechanism, resulting in a higher frequency of oedema.
Are there any evidence showing the increased activity of the immune mechanisms involving complement in early stage compared to late stage cancer?
Answer Only in our study, we observed particularly low values of C1INH and C4 in the early phase of breast cancer, which may be dictated by their consumption mechanism in the stage of the rapid growth of the lymphatic tissue. However, this requires further research, including the exclusion of autoantibody involvement.
In other cancer such as colorectal and lung cancer, are C1-INH-AAEs more frequent in early stage?
Answer We did not observe the such an event.
Why is BMI associated with occurrence of C1-INH-AAEs in breast cancer? Is BMI also associated with occurrence of C1-INH-AAEs in other cancer except for breast cancer?
Answer There are some observations that BMI is associated with the occurrence of acquired angioedema. However, there is no simple explanation for this phenomenon. Possibly linking overweight with the event of other metabolic disorders, e.g. hyperuricemia, increases the risk of angioedema (Bozek A, Zajac M Ann Allergy 2011)This relationship has not yet been studied in detail, but it is an exciting topic for further exploration.
It has been shown that the risk of breast cancer increases by 12% for every 5 kg/m2 above average body weight. In women with a BMI over 28 kg/m2, the risk increases by 26%. In addition, obesity is a modifiable risk factor for developing breast cancer. In the presented study, most women with breast cancer had elevated BMI, especially in the early phase of the disease. This may indirectly further explain the relationship between angioedema and BMI.
(Mazur-Roszak M., Litwiniuk M., Grodecka-Gazdecka
- OtyÅ‚ość a rak piersi. WspóÅ‚. Onkol. 2010; 14 (4): 270–275)
Sinicrope F.A., Dannenberg A.J. Obesity and
breast cancer prognosis: weight of the evidence.J. Clin. Oncol. 2011; 29 (1): 4–7.)
Thank you very much for your valuable comments
Reviewer 2 Report
This is a paper presenting data about the prevalence of aquired angioedema with C1 INH deficiency in patients with cancer. The idea is interesting. This is a rare disease, and we need to know more and more about it.
But I have many doubts about the research design, and consequently about its results. In my opinion it is not clear if the populations described (both the patients with and without neoplasms) are collected based on suspect of angioedema or not. In case they are not why they have so many patients that have done the dosage of C1 inh? it seens a very high number of patients, considering that this is a very rare disease, and the dosage of C1 inh is performed very rarely. Moreover the criteria used to reach a iagnosis of AAE-C! inh are not clear.In tabel 1 the authors say that 127 patients had 1 episode of angioedema, 103 had2 2 or 3 episodes, and just in 97 patients the angioedema is recurrent. so the question is....even patients with one episode of angioedema the medical doctor decided to perform the dosage of C1 inhibitor? And was reached a diagnosis of AAE-C1 INH?
I think it is important to clarify better the design of the study, and also the criteria used for the diagnosis. It is possible that the 327 patients with neoplasm and angioedema are not all AAE-1 inh? the number is very high based on the rarity of the disease. and it is very strange that the diagnosis was perfomed in patients with a single angioedema attack. It is probable that I have not understood some step of the paper.
Another point where i think it is important to clarify the details if the figure 1. this is the serum concentration in patients with cancer? or in patients with cancer and AAE-C1 inh? the same question is valid also for fogure 2 and 3.
Did you look for antibodies anti-C1 inh in these patints? it is an interesting point to discuss.
Line 143. The authors say that acquired angioedema is a common diesease symtom. Why common? Itis important to clarify.
Line 144. the authors say...another area of improvement....it is not clear. improvement of whtìat? it is useful to clarify.
line 152: the authors say this first event was observed. which event? the authors demonstrated that there are antibodies antic1inh in these patients?
Line 189: did the authors perofmr mutation assessment in this patients?
Line 198: it is not clear.....the angioedema is connected to the use of drugs? in this case this is not a aae-c1inh.
Author Response
I think it is important to clarify better the design of the study, and also the criteria used for the diagnosis. It is possible that the 327 patients with neoplasm and angioedema are not all AAE-1 inh? the number is very high based on the rarity of the disease. and it is very strange that the diagnosis was perfomed in patients with a single angioedema attack. It is probable that I have not understood some step of the paper.
Answer: Indeed, some points of the study protocol were not adequately presented. This has been specified. All included cancer patients and the control group were suspected of angioedema. For this reason, ALL had serum C1inh. Our observation concerned only patients with reduced C1 inhibitory values. Previously, it was not known to what extent this angioedema occurs. For this reason, our data try to specify this with restrictive criteria excluding other types of angioedema that have occurred there. We have included all doubts in the limitations of work. The follow-up time was too short to assess the true amount of angioedema recurrence. However, even single incidents of such edema can be classified as C1INHAAE according to literature data (1,5)
Another point where i think it is important to clarify the details if the figure 1. this is the serum concentration in patients with cancer? or in patients with cancer and AAE-C1 inh? the same question is valid also for figure 2 and 3.
Answer: All presented patients have C1-INH-AAE in figure 1,2 and 3 .It was clarified
Did you look for antibodies anti-C1 inh in these patients? it is an interesting point to discuss.
Answer: Unfortunately, we did not. It was mentioned in limitation of the study
Line 143. The authors say that acquired angioedema is a common diesease symtom. Why common? Itis important to clarify.
Answer: This sentence was changed according literature data.
Line 144. the authors say...another area of improvement....it is not clear. improvement of whtìat? it is useful to clarify.
Answer: This sentence was deleted
line 152: the authors say this first event was observed. which event? the authors demonstrated that there are antibodies antic1inh in these patients?
Answer: It was corrected. It was first such extend observation about C1INHAAE in cancer diseases
Line 189: did the authors perofmr mutation assessment in this patients?
Answer: We did not.
Line 198: it is not clear.....the angioedema is connected to the use of drugs? in this case this is not a aae-c1inh.
Answer: It was clarified in methods and the last sentence was removed.
Reviewer 3 Report
1. why is the age classification in this study starting from 35 years? if in the introduction it is stated that the onset of angioedema is over 40 years why not start from 40 years?
2. Is a history of angioedema with anaphylactic shock included in the inclusion criteria?
Author Response
Why is the age classification in this study starting from 35 years? if in the introduction it is stated that the onset of angioedema is over 40 years why not start from 40 years?
Answer: We lowered the age of the respondents from 40 to 35 because, from this lower age, women most often suffer from breast cancer in Poland. We wanted to check whether there are also cases of the studied type of angioedema in this youngest group. Our data confirm this, but the frequency is not dependent on the examined age (+35). In the literature, the age of 40 years for C1INH-AEE type angioedema is approximate, and there are no literature data precisely specifying this limit, although it is sometimes placed.
Is a history of angioedema with anaphylactic shock included in the inclusion criteria?
Yes, however there were no such cases.
Round 2
Reviewer 1 Report
I have no more comments.
Reviewer 2 Report
ok